# Diffusion Models for Video Prediction and Infilling

**Tobias Höppe**                                                        *tobihoeppe@gmail.com*
*KTH Stockholm*

**Arash Mehrjou**[*]
*MPI for Intelligent Systems & ETH Zürich*

**Stefan Bauer**[*]
*KTH Stockholm*

**Didrik Nielsen**[*]
*Norwegian Computing Center*

**Andrea Dittadi**[*]                                                        *adit@dtu.dk*
*Technical University of Denmark & MPI for Intelligent Systems*

**Reviewed on OpenReview:** *https://openreview.net/forum?id=1f0lr4AYM6*

## Abstract

Predicting and anticipating future outcomes or reasoning about missing information in a sequence are critical skills for agents to be able to make intelligent decisions. This requires strong, temporally coherent generative capabilities. Diffusion models have shown remarkable success in several generative tasks, but have not been extensively explored in the video domain. We present Random-Mask Video Diffusion (RaMViD), which extends image diffusion models to videos using 3D convolutions, and introduces a new conditioning technique during training. By varying the mask we condition on, the model is able to perform video prediction, infilling, and upsampling. Due to our simple conditioning scheme, we can utilize the same architecture as used for unconditional training, which allows us to train the model in a conditional and unconditional fashion at the same time. We evaluate RaMViD on two benchmark datasets for video prediction, on which we achieve state-of-the-art results, and one for video generation. High-resolution videos are provided at https://sites.google.com/view/video-diffusion-prediction.

## 1 Introduction

Videos contain rich information about the world, and a vast amount of diverse video data is available. Training models that understand this data can be crucial for developing agents that interact with the surrounding world effectively. In particular, video prediction plays an increasingly important role: autonomous driving (Hu et al., 2020), anticipating events (Zeng et al., 2017), planning (Finn & Levine, 2017) and reinforcement learning (Hafner et al., 2019) are applications which can benefit from increasing performance of prediction models. On the other hand, video infilling—i.e., observing a part of a video and generating missing frames—can be used for example in planning, estimating trajectories, and video processing. In addition, video models can be valuable for downstream tasks such as action recognition (Kong & Fu, 2018) and pose estimation (Sahin et al., 2020). However, there has not been extensive research on video infilling and most research is focusing on generation or prediction.

Most recent approaches to video prediction are based on variational autoencoders (Babaeizadeh et al., 2021; Saxena et al., 2021) or GANs (Clark et al., 2019; Luc et al., 2020). Diffusion models (Sohl-Dickstein et al.,

---

[*]Equal advising.

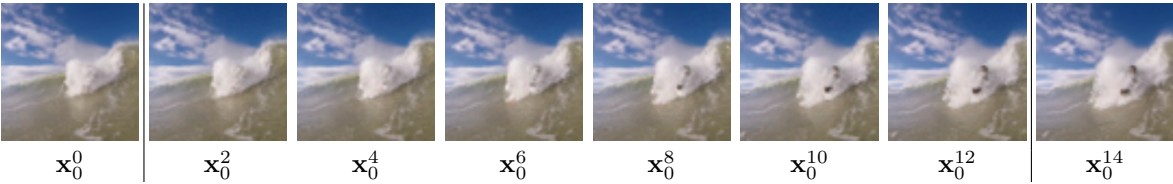

$$\mathbf{x}_0^0 \qquad \mathbf{x}_0^2 \qquad \mathbf{x}_0^4 \qquad \mathbf{x}_0^6 \qquad \mathbf{x}_0^8 \qquad \mathbf{x}_0^{10} \qquad \mathbf{x}_0^{12} \qquad \mathbf{x}_0^{14}$$

Figure 1: The first two and last two frames of a video are given and our model does fill in the missing frames very accurate and with much detail.

2015; Ho et al., 2020; Nichol & Dhariwal, 2021; Song et al., 2021; Abstreiter et al., 2021; Mittal et al., 2022; Dockhorn et al., 2021) have recently seen tremendous progress on static visual data, even outperforming GANs in image synthesis (Dhariwal & Nichol, 2021), but have not yet been extensively studied for videos. Considering their impressive performance on images, it is reasonable to believe that diffusion models may also be useful for tasks in the video domain.

In this paper, we extend diffusion models to the video domain via several technical contributions. We use 3D convolutions and a new conditioning procedure incorporating randomness. Our model is not only able to predict future frames of a video but also fill in missing frames at arbitrary positions in the sequence (see Fig. 1). Therefore, our Random-Mask Video Diffusion (RaMViD) can be used for several video completion tasks. We summarize our technical contributions as follows:[1]

- A novel diffusion-based architecture for video prediction and infilling.
- Competitive performance with recent approaches across multiple datasets.
- Introduce a schedule for the random masking.

The remainder of this paper is organized as follows: In Section 2, we provide the necessary background on diffusion models and video prediction and outline relevant related work. Section 3 describes Random-Mask Video Diffusion (RaMViD). In Section 4, we present and discuss extensive experiments on several benchmark datasets. We finally conclude with a discussion in Section 5.

## 2 Background and related work

**Diffusion models.** Diffusion-based models generally refer to the class of machine learning algorithms that consist of gradually transforming a complex distribution into unstructured noise and learning to reverse this process to recover the data generating distribution. They have attracted a great deal of attention after being successfully applied to a diverse range of tasks such as image generation (Song et al., 2021; Niu et al., 2020), audio (Chen et al., 2021), graph and shape generation (Cai et al., 2020). The essence of these models is two stochastic (diffusion) processes implemented by Stochastic Differential Equations (SDEs), a forward and a backward one. We explain the formulation in the abstract domain here and specialize it later according to the application of this work.

Let $\mathbf{x}_0 \in \mathbb{R}^d$ be a sample from the empirical data distribution, i.e., $\mathbf{x}_0 \sim p_{\text{data}}(\mathbf{x}_0)$ and $d$ be the data dimension. The forward diffusion process takes $\mathbf{x}_0$ as the starting point and creates the random trajectory $\mathbf{x}_{[0,T]}$ from $t = 0$ to the final time $t = T$. The forward process is designed such that $p(\mathbf{x}_T \,|\, \mathbf{x}_0)$ has a simple unstructured distribution. One example of such SDEs is

$$d\mathbf{x}_t = f(\mathbf{x}_t, t)dt + g(t)dw := \sqrt{\frac{d[\sigma^2(t)]}{dt}}dw \;, \tag{1}$$

where $w$ is the Brownian motion. A desirable property of this process is the fact that the conditional distribution $p(\mathbf{x}_t \,|\, \mathbf{x}_0)$ takes a simple analytical form:

$$p(\mathbf{x}_t \,|\, \mathbf{x}_0) = \mathcal{N}\left(\mathbf{x}_t; \mathbf{x}_0, \left(\sigma^2(t) - \sigma^2(0)\right)\mathbf{I}\right). \tag{2}$$

---

[1]Code is available at `https://github.com/Tobi-r9/RaMViD`.

Upon learning the gradient of $p(\mathbf{x}_t)$ for each $t$, one can reverse the above process and obtain the complex data distribution from pure noise as

$$d\mathbf{x}_t = [f(\mathbf{x}_t, t) - g^2(t)\nabla_{\mathbf{x}} \log p(\mathbf{x}_t)]dt + g(t)dw' \, , \tag{3}$$

where $w'$ is a Brownian motion independent of the one in the forward direction. Hence, generating samples from the data distribution boils down to learning $\nabla_{\mathbf{x}} \log p(\mathbf{x})$.

The original score matching objective (Hyvärinen & Dayan, 2005):

$$\mathbb{E}_{\mathbf{x}_t} \left[ \|s_\theta(\mathbf{x}_t, t) - \nabla_{\mathbf{x}_t} \log p(\mathbf{x}_t)\|_2^2 \right] \tag{4}$$

is the most intuitive way to learn the score function, but is unfortunately intractable. Denoising Score Matching (DSM) provides a tractable alternative objective function:

$$J_t^{DSM}(\theta) = \mathbb{E}_{\mathbf{x}_0}\mathbb{E}_{\mathbf{x}_t|\mathbf{x}_0} \left[ \|s_\theta(\mathbf{x}_t, t) - \nabla_{\mathbf{x}_t} \log p(\mathbf{x}_t \,|\, \mathbf{x}_0)\|_2^2 \right] \tag{5}$$

whose equivalence with the original score matching objective was shown by Vincent (2011) and used to train energy models by Saremi et al. (2018). Similarly to many recent works, we use the DSM formulation of score matching to learn the score function.

**Video prediction and infilling.** Research in video prediction has received more attention in the previous years, as the ability to predict videos can be used for several downstream tasks (Oprea et al., 2020). Video prediction can be modeled in a deterministic or stochastic form. Deterministic modeling (Walker et al., 2015; Vondrick & Torralba, 2017; Terwilliger et al., 2019; Sun et al., 2019) tries to predict the most likely future, but this often leads to averaging the future states (Li et al., 2019). Due to the stochastic nature of the future, generative models have lately shown to be more successful in capturing the underlying dynamics. For this approach, variational models are often used by modeling the stochastic content in a latent variable (Babaeizadeh et al., 2018; Saxena et al., 2021; Denton & Fergus, 2018; Wu et al., 2021a). However, this often leads to blurry predictions due to underfitting, and Babaeizadeh et al. (2021) have overcome this problem via architectural novelties. Blurry prediction is a less serious problem in GANs and promising results have been achieved especially on large datasets (Clark et al., 2019; Luc et al., 2020). On the other hand, the body of research on video infilling is significantly more scarce, with most works in this area focusing on frame interpolation (Jiang et al., 2018). However, Xu et al. (2018) have shown interesting results in infilling, by modeling the video as a stochastic generation process.

**Concurrent work.** Yang et al. (2022) is the only work so far that has used diffusion models for autoregressive video prediction, by modeling residuals for a predicted frame. However, since their evaluation procedure and datasets are different, a comparison with their work is not possible. A few concurrent works have recently considered diffusion models for video generation. Ho et al. (2022) focus on unconditional video generation, Harvey et al. (2022) use diffusion models to predict long videos, and Voleti et al. (2022), the most similar to our work, also consider video prediction and infilling.

## 3 Random-Mask Video Diffusion

Our method, Random-Mask Video Diffusion (RaMViD), consists of two main features. First, the way we introduce conditional information is different from prior work. Second, by randomizing the mask, we can directly use the same approach for video prediction and video completion (infilling). In the following, we detail each of these aspects of RaMViD.

### 3.1 Conditional training

Let $\mathbf{x}_0 \in \mathbb{R}^{L,W,H,C}$ be a video with length $L$. We partition the video $\mathbf{x}_0$ into two parts: the unknown frames $\mathbf{x}_0^{\mathcal{U}} \in \mathbb{R}^{L-k,W,H,C}$ and the conditioning frames $\mathbf{x}_0^{\mathcal{C}} \in \mathbb{R}^{k,W,H,C}$, where $\mathcal{U}$ and $\mathcal{C}$ are sets of indices such that $\mathcal{U} \cap \mathcal{C} = \varnothing$ and $\mathcal{U} \cup \mathcal{C} = \{0, 1, \dots, L-1\}$. We write $\mathbf{x}_0 = \mathbf{x}_0^{\mathcal{U}} \oplus \mathbf{x}_0^{\mathcal{C}}$ with the following definition for $\oplus$:

$$(\mathbf{a}^{\mathcal{U}} \oplus \mathbf{b}^{\mathcal{C}})^i := \begin{cases} \mathbf{a}^i \text{ if } i \in \mathcal{U} \\ \mathbf{b}^i \text{ if } i \in \mathcal{C} \end{cases} \tag{6}$$

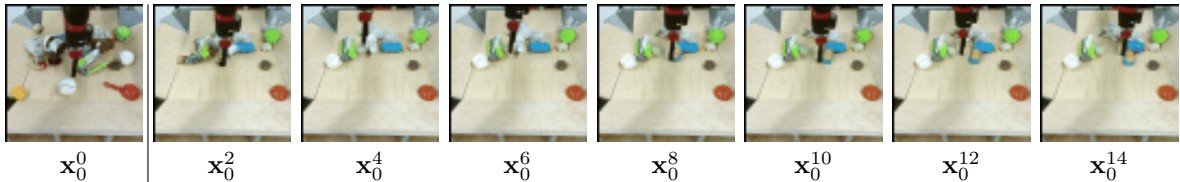

$$\mathbf{x}_0^0 \qquad \mathbf{x}_0^2 \qquad \mathbf{x}_0^4 \qquad \mathbf{x}_0^6 \qquad \mathbf{x}_0^8 \qquad \mathbf{x}_0^{10} \qquad \mathbf{x}_0^{12} \qquad \mathbf{x}_0^{14}$$

Figure 2: An unconditionally trained model is used to predict 15 frames given one frame. Even with re-sampling we can see, that objects in the background are not harmonized between the predicted and conditioned frames.

where the superscript $i$ indicates tensor indexing and in our case corresponds to selecting a frame from a video. Here, $t$ indicates the diffusion step, with $t = 0$ corresponding to the data and $t = T$ to the prior Gaussian distribution.

If we use an unconditionally trained model, we find that the predicted unknown frames $\mathbf{x}_0^{\mathcal{U}}$ do not harmonize well with the conditioning frames $\mathbf{x}_0^{\mathcal{C}}$, as shown in Fig. 2. One solution for this would be re-sampling, as proposed by Lugmayr et al. (2022). In re-sampling, we take one step in the learned reversed diffusion (denoising) process and then go back by taking a step in the forward diffusion process (i.e., adding noise again). This is repeated several times for each diffusion step, to make sure the predicted and conditioning frames are harmonized. However, this becomes computationally too expensive for videos, especially when using very few conditioning frames, as the number of re-sampling steps needs to be increased. To mitigate this issue, we propose to train the model conditionally with *randomized masking*.

Conditional diffusion models usually optimize

$$\mathbb{E}_{\mathbf{x}_0} \left\{ \mathbb{E}_{\mathbf{x}_t | \mathbf{x}_0} \left[ \left\| s_\theta(\mathbf{x}_t, \mathbf{x}_0^{\mathcal{C}}, t) - \nabla_{\mathbf{x}_t} \log p(\mathbf{x}_t | \mathbf{x}_0) \right\|_2^2 \right] \right\} \tag{7}$$

where $\mathbf{x}_0^{\mathcal{C}}$ is typically given as a separate input through an additional layer (Chen et al., 2021) or it is concatenated with the input (Saharia et al., 2021b; Batzolis et al., 2021; Saharia et al., 2021a). On the other hand, we feed the entire sequence to the network $s_\theta$ but only add noise to the unmasked frames: $\mathbf{x}_0^{\mathcal{U}} \sim \mathcal{N}\left(\mathbf{x}_0^{\mathcal{U}}, \left(\sigma^2(t) - \sigma^2(0)\right)\mathbf{I}\right)$. The input to the network is then a video where some frames are noisy and some are clean: $\mathbf{x}_t = \mathbf{x}_t^{\mathcal{U}} \oplus \mathbf{x}_0^{\mathcal{C}}$ (see Fig. 3). The loss is computed only with respect to $\mathbf{x}_t^{\mathcal{U}}$:

$$J_t^{\mathrm{RaMViD}}(\theta) = \mathbb{E}_{\mathbf{x}_0} \left\{ \mathbb{E}_{\mathbf{x}_t^{\mathcal{U}} | \mathbf{x}_0} \left[ \left\| s_\theta(\mathbf{x}_t, t)^{\mathcal{U}} - \nabla_{\mathbf{x}_t^{\mathcal{U}}} \log p(\mathbf{x}_t^{\mathcal{U}} | \mathbf{x}_0) \right\|_2^2 \right] \right\}. \tag{8}$$

where $s_\theta(\mathbf{x}_t, t)^{\mathcal{U}}$ represents the output of the model with indices in $\mathcal{U}$. Note that the score function $\nabla_{\mathbf{x}_t^{\mathcal{U}}} \log p(\mathbf{x}_t^{\mathcal{U}} | \mathbf{x}_0)$ has the same dimension as $\mathbf{x}_t^{\mathcal{U}}$, whereas in Eq. (7) it has the dimension of the entire video $\mathbf{x}_t$. This leads to the forward diffusion process:

$$d\mathbf{x}_t^{\mathcal{U}} = f(\mathbf{x}_t^{\mathcal{U}}, t)dt + g(t)dw \tag{9}$$

and the reversed diffusion process then becomes:

$$d\mathbf{x}_t^{\mathcal{U}} = [f(\mathbf{x}_t^{\mathcal{U}}, t) - g^2(t)\nabla_{\mathbf{x}_t^{\mathcal{U}}} \log p(\mathbf{x}_t^{\mathcal{U}} | \mathbf{x}_0^{\mathcal{C}})]dt + g(t)dw' . \tag{10}$$

Similarly, Tashiro et al. (2021) compute the loss only on the unknown input. However, they also use concatenation and zero-padding to bring $\mathbf{x}_t^{\mathcal{C}}$ and $\mathbf{x}_t$ to the same dimension. For a more detailed schematic, see Appendix A. In our implementation, we used a discrete diffusion process with $t \in \{0, 1, \ldots, T-1, T\}$.

## 3.2 Randomization

As previously mentioned, the proposed model is able to perform several tasks. We achieve this by sampling $\mathcal{C}$ at random. At each training step, we first choose the number of conditioning frames $|\mathcal{C}| = k \in \{1, \ldots, K\}$, where $K$ is a chosen hyperparameter. Then we define $\mathcal{C}$ by selecting $k$ random indices from $\{0, \ldots, L-1\}$, and we refrain from applying the diffusion process to the corresponding frames. Since the videos now consist

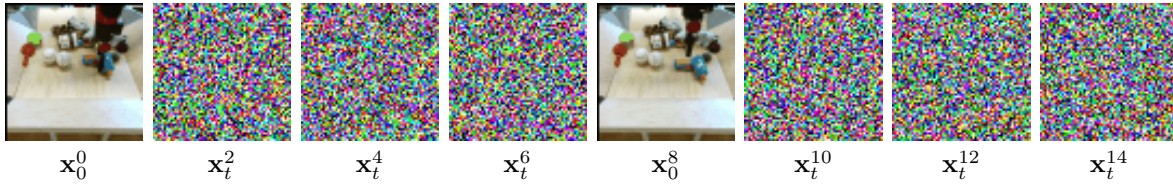

$$\mathbf{x}_0^0 \qquad \mathbf{x}_t^2 \qquad \mathbf{x}_t^4 \qquad \mathbf{x}_t^6 \qquad \mathbf{x}_0^8 \qquad \mathbf{x}_t^{10} \qquad \mathbf{x}_t^{12} \qquad \mathbf{x}_t^{14}$$

Figure 3: Example input of the network with $\mathcal{C} = \{0, 8\}$.

of original and noisy frames in varying positions, the model has to learn to distinguish between them in order to use the frames $\mathbf{x}_0^{\mathcal{C}}$ as information for the reversed diffusion process. After training, we can use RaMViD by fixing $\mathcal{C}$ to the set of indices of the known frames ($\mathcal{C}$ can be any arbitrary subset of $\{0, \ldots, L-1\}$) and generating the unknown frames (those with indices in $\mathcal{U}$).

Our approach allows us to use the exact same architecture of unconditionally trained models, thus enabling *mixed training*, where we train the model conditionally and unconditionally at the same time. We set $\mathcal{C} = \varnothing$ (i.e., the model does not have any conditional information $\mathbf{x}_t^{\mathcal{C}}$) with probability $p_U$, which is a fixed hyperparameter. If $\mathcal{C} = \varnothing$, our objective in Eq. (8) becomes the same as the objective in Eq. (5) used for unconditional training. The pseudocode for RaMViD is shown in Algorithm 1.

---

**Algorithm 1** RaMViD.

---
Initialize model $\sim s_\theta$
$T$ = Number of diffusion steps
$K$ = Max number of frames to condition on
$L$ = Length of the video
**while** not converged **do**
    $\mathbf{x}_0 \sim p_{\text{data}}(\mathbf{x}_0)$
    $t \sim \text{Uniform}(\{0, \ldots, T\})$
    $b \sim \text{Bernoulli}(p_U)$
    **if** $b$ **then**
        $\mathcal{C} = \varnothing$
    **else**
        $k \sim \text{Uniform}(\{1, \ldots, K\})$
        $\mathcal{C} \sim \text{Uniform}\left(\{S \subseteq \{0, \ldots, L-1\} : |S| = k\}\right)$
    **end if**
    $\mathcal{U} = \{0, \ldots, L-1\} \setminus \mathcal{C}$
    $\mathbf{x}_t^{\mathcal{U}} \sim \mathcal{N}\left(\mathbf{x}_t^{\mathcal{U}}; \mathbf{x}_0^{\mathcal{U}}, \left(\sigma^2(t) - \sigma^2(0)\right)\mathbf{I}\right)$
    $\mathbf{x}_t = \mathbf{x}_t^{\mathcal{U}} \oplus \mathbf{x}_0^{\mathcal{C}}$
    Take a gradient step on: $\nabla_\theta \mathbb{E}_{\mathbf{x}_0}\left\{\mathbb{E}_{\mathbf{x}_t^{\mathcal{U}}|\mathbf{x}_0}\left[\left\|s_\theta(\mathbf{x}_t, t)^{\mathcal{U}} - \nabla_{\mathbf{x}_t^{\mathcal{U}}}\log p(\mathbf{x}_t^{\mathcal{U}}|\mathbf{x}_0)\right\|_2^2\right]\right\}$
**end while**

---

## 4 Experiments

### 4.1 Experimental setup

**Implementation details.** Our implementation relies on the official code of Nichol & Dhariwal (2021),[2] adapted to video data by using 3D convolutions. Even though most previous work uses the cosine noise schedule, we found that the linear noise schedule works better when training the model conditionally. Therefore, we use a linear diffusion schedule for our experiments. For the architecture, we also use the same as proposed by Nichol & Dhariwal (2021): a U-Net with self-attention at the resolutions 16 and 8. We do not encode the time dimension. We use two ResNet blocks per resolution for the BAIR dataset, and three blocks

---

[2]https://github.com/openai/improved-diffusion

for Kinetics-600 and UCF-101. We set the learning rate for all our experiments to 2e-5, use a batch size of 32 for BAIR and 64 for Kinetics-600 and UCF-101, and fix $T = 1000$. We found, especially on the more diverse datasets like Kinetics-600 and UCF-101, that larger batch sizes produce better results. Therefore, to increase the batch size, we use gradient accumulation by computing the gradients for micro-batches of size 2 and accumulate for several steps before doing back-propagation.

**Datasets and evaluations.** To compare our model to prior work, we train it on the BAIR robot pushing dataset (Ebert et al., 2017). The dataset consists of short videos, with $64 \times 64$ resolution, of a robot arm manipulating different objects. For evaluation, we use the same setting as Rakhimov et al. (2020), which is to predict the next 15 frames given one observed frame. We train on videos of length 20.

Additionally, we evaluate our model on the Kinetics-600 dataset (Carreira et al., 2018), which consists of roughly 500,000 10-second YouTube clips, also at $64 \times 64$ resolution, from 600 classes. The size and the diversity of this dataset make it a perfect task to investigate if the model captures the underlying real-world dynamics. For downloading and preprocessing we use the dataset's public repository.[3] On Kinetics-600, we compare our model to concurrent work by predicting 11 frames when conditioned on 5 frames (Luc et al., 2020). We additionally perform several ablation studies on video completion. We train on 16 frames and choose again $K = 4$.

To quantitatively evaluate the unconditional generation performance when using $p_U > 0$, we also train on UCF-101 (Soomro et al., 2012), a common benchmark for unconditional video generation. It consists of 13,320 videos from 101 human action classes. We also rescale this dataset to $64 \times 64$ and train with $K = 4$.

To quantitatively evaluate prediction, we use the Fréchet Video Distance (FVD) (Unterthiner et al., 2018),[4] which captures semantic similarity and temporal coherence between videos by comparing statistics in the latent space of a Inflated 3D ConvNet (I3D) trained on Kinetics-400. To evaluate unconditional generation, we use the Inception Score (IS) (Salimans et al., 2016) to measure the quality and diversity of the generated videos. As we have to adapt the score to videos, we use the public repository from Saito et al. (2020).[5]

## 4.2 BAIR

We train four models on the BAIR dataset with $p_U \in \{0, 0.25, 0.5, 0.75\}$ respectively. The models are trained for 250,000 iterations with a batch size of 32 on 8 GPUs.

First, we test our method with the typical evaluation protocol for BAIR (predicting 15 frames, given one conditional frame). With all values of $p_U$, we can achieve state-of-the-art performance, as shown in Table 1. By using $p_U > 0$, we can even increase the performance of our method. However, it seems that there is a tipping point after which the increasing unconditional rate hurts the prediction performance of the model. Interestingly, we find that also the model trained with $p_U = 0.75$ overcomes the harmonization problem described in Section 3.1. We have trained more models with $p_U = 0.25$ but varying $K$. When training with $K = 2$ we found a slight drop in performance but with $K = 8$ a slight increase. Furthermore, we experiment with a task-specific model for prediction (i.e. setting $\mathcal{C} = 0$ for all training steps), which we will call *RaMViD fixed*. However, we find that this does not improve performance and does not appear to work reliably on other video completion tasks.

Since we train with randomized masking, we can also perform video infilling with the same models, without retraining. We condition on the first and last frame (i.e., set $\mathcal{C} = \{0, 15\}$ for sampling) and compute the FVD of the 14 generated frames. Again we find that the performance is very similar for different values of $p_U$ (see Table 2), and only a slight change in results when using different values for $K$. Interestingly, RaMViD is also able to perform unconditional generation on BAIR for all considered values of $p_U$, as shown in Appendix B.1.

So far, we have shown that our method works very well for prediction and infilling. However, since the BAIR dataset is arguably rather simple and not very diverse, we will now evaluate RaMViD on the significantly

---

[3]https://github.com/cvdfoundation/kinetics-dataset
[4]https://github.com/google-research/google-research/tree/master/frechet_video_distance
[5]https://github.com/pfnet-research/tgan2

Table 1: Prediction performance on BAIR. The values are taken from Babaeizadeh et al. (2021) after inquiring about the evaluation procedure. Parameter counts were obtained either directly from the papers or by contacting the authors. Since our computational constraints did not allow us to do several runs for each method, we only give error bars for RaMViD ($p_U = 0.25$)

| Method | FVD ($\downarrow$) | # parameters |
|---|---|---|
| Latent Video Transformer (Rakhimov et al., 2020) | 125.8 | |
| SAVP (Lee et al., 2018) | 116.4 | |
| DVD-GAN-FP (Clark et al., 2019) | 109.8 | |
| TrIVD-GAN-FP (Luc et al., 2020) | 103.3 | |
| VideoGPT (Yan et al., 2021) | 103.3 | 40M |
| Video Transfomer (Weissenborn et al., 2020) | 94.0 | 373M |
| FitVid (Babaeizadeh et al., 2021) | 93.6 | 302M |
| MCVD (Voleti et al., 2022) | 89.5 | 251.2M |
| NÜWA (Wu et al., 2021b) | 86.9 | |
| RaMViD ($p_U = 0, K = 4$) | 86.41 | 235M |
| RaMViD ($p_U = 0.25, K = 4$) | $85.3 \pm 1.8$ | 235M |
| RaMViD ($p_U = 0.5, K = 4$) | 85.03 | 235M |
| RaMViD ($p_U = 0.75, K = 4$) | 86.05 | 235M |
| RaMViD ($p_U = 0.25, K = 2$) | 87.39 | 235M |
| RaMViD ($p_U = 0.25, K = 8$) | **82.64** | 235M |
| RaMViD ($p_U = 0.25$, *fixed*) | 89.01 | 235M |

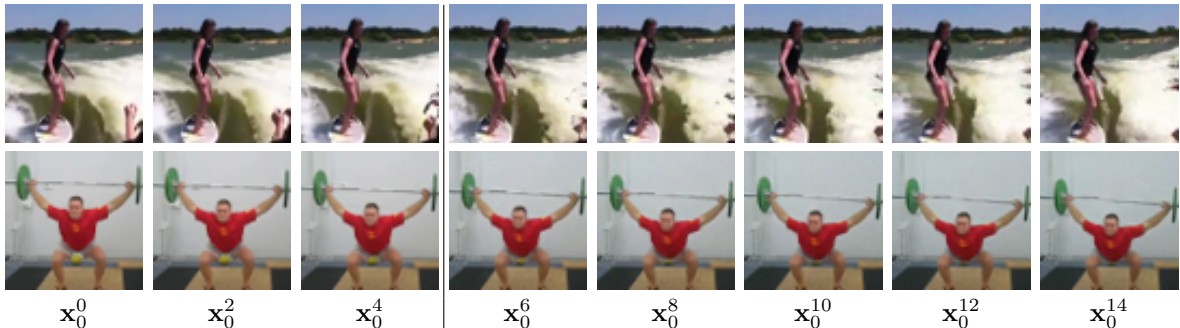

$\mathbf{x}_0^0 \qquad \mathbf{x}_0^2 \qquad \mathbf{x}_0^4 \qquad \mathbf{x}_0^6 \qquad \mathbf{x}_0^8 \qquad \mathbf{x}_0^{10} \qquad \mathbf{x}_0^{12} \qquad \mathbf{x}_0^{14}$

Figure 4: Prediction of 11 frames given the first 5 frames on Kinetics-600 with RaMViD ($p_U = 0.25$).

more complex Kinetics-600 dataset. Since using $K = 4$ appears to lead to stable performance on all tasks we evaluated our model on, we will use $K = 4$ for our experiments on Kinetics-600 and UCF-101.

Table 2: Infilling performance on BAIR.

| Method | FVD ($\downarrow$) |
|---|---|
| RaMViD ($p_U = 0, K = 4$) | 85.68 |
| RaMViD ($p_U = 0.25, K = 4$) | 85.02 |
| RaMViD ($p_U = 0.5, K = 4$) | 87.04 |
| RaMViD ($p_U = 0.75, K = 4$) | 87.85 |
| RaMViD ($p_U = 0.25, K = 2$) | **83.83** |
| RaMViD ($p_U = 0.25, K = 8$) | 87.11 |
| RaMViD ($p_U = 0.25$, *fixed*) | 119.76 |

Table 3: Prediction performance on Kinetics-600. Values are taken from Moing et al. (2021) after inquiring about the evaluation procedure. Parameter counts were obtained either directly from the papers or by contacting the authors. Since our computational constraints did not allow us to do several runs for each method, we only give error bars for RaMViD ($p_U = 0.25$)

| Method | FVD ($\downarrow$) | # parameters |
|---|---|---|
| Video Transfomer (Weissenborn et al., 2020) | $170 \pm 5$ | 373M |
| DVD-GAN-FP (Clark et al., 2019) | $69 \pm 1$ | |
| CCVS (Moing et al., 2021) | $55 \pm 1$ | 366M |
| TrIVD-GAN-FP (Luc et al., 2020) | $26 \pm 1$ | |
| RaMViD ($p_U = 0$) | 18.69 | 308M |
| RaMViD ($p_U = 0.25$) | $\mathbf{17.53} \pm 1.07$ | 308M |
| RaMViD ($p_U = 0.5$) | 17.61 | 308M |
| RaMViD ($p_U = 0.75$) | 27.64 | 308M |

Table 4: Performance of RaMViD on Kinetics-600, when conditioning on different frames.

| Method | $\mathcal{C} = \{0, 1, 14, 15\}$ | $\mathcal{C} = \{0, 5, 10, 15\}$ |
|---|---|---|
| RaMViD ($p_U = 0$) | **10.68** | 6.28 |
| RaMViD ($p_U = 0.25$) | 10.85 | **4.91** |
| RaMViD ($p_U = 0.5$) | 10.86 | 5.90 |
| RaMViD ($p_U = 0.75$) | 17.33 | 7.29 |

## 4.3 Kinetics-600

For the Kinetics-600 dataset, we increase the batch size to 64 and train for 500,000 iterations on 8 GPUs. First, we evaluate the model on prediction with the setting described in Section 4.1 (predict 11 frames given 5 frames). When comparing our models to concurrent work, we find that RaMViD achieves state-of-the-art results by a significant margin (see Table 3). In Fig. 4, we can see that the model produces temporally coherent outputs and is able to model details, especially in the background, such as clouds and patterns in the water. Nevertheless, it struggles with fast movements: objects moving quickly often get deformed, as can be observed in Appendix B.2. Similar to what we have seen in Table 1, having an unconditional rate $p_U > 0$ increases the performance up to a tipping point. However, differently from the model trained on BAIR, the FVD score now drops significantly with $p_U = 0.75$. We conjecture that this drop in performance is due to the complexity of the data distribution. In BAIR, the conditional and unconditional distributions are rather similar, while this is not true for Kinetics-600.

We also evaluate RaMViD on two video completion tasks on Kinetics-600. The first task is to fill in a video given the two first and last frames (i.e., $\mathcal{C} = \{0, 1, 14, 15\}$): the challenge here is to harmonize the observed movement at the beginning with the movement observed at the end. In the second task, the conditioning frames are distributed evenly over the sequence (i.e., $\mathcal{C} = \{0, 5, 10, 15\}$), hence the model has to infer the movement from the static frames and harmonize them into one realistic video. RaMViD excels on both tasks, as shown quantitatively in Table 4 and qualitatively in Figs. 11 and 12. Especially when setting $\mathcal{C} = \{0, 5, 10, 15\}$ RaMViD is able to fill the missing frames with very high quality and coherence. This setting can be easily applied to upsampling by training a model on high-FPS videos and then sampling a sequence conditioned on a low-FPS video.

We find that only RaMViD ($p_U = 0.5$) and RaMViD ($p_U = 0.75$) can generate unconditional videos on Kinetics-600. To quantify RaMViD's unconditional generation, we will evaluate these models on the UCF-101 dataset and compare it to other work.

Table 5: Generative performance of RaMViD on UCF-101. Note that the methods TGAN-F, VideoGPT and DVD-GAN in Table 5 are trained with $128 \times 128$ resolution, which gives them a slight advantage, as the IS score is computed with $112 \times 112$ resolution.

| Method | IS ($\uparrow$) | # parameters | resolution |
|---|---|---|---|
| VGAN (Vondrick et al., 2016) | $8.31 \pm 0.09$ | | $64 \times 64$ |
| MoCoGAN (Tulyakov et al., 2018) | $12.42 \pm 0.03$ | 3.3M | $64 \times 64$ |
| TGAN-F (Kahembwe & Ramamoorthy, 2020) | $13.62 \pm 0.06$ | 17.5M | $64 \times 64$ |
| progressive VGAN (Acharya et al., 2018) | $14.56 \pm 0.05$ | | $64 \times 64$ |
| TGAN-F (Kahembwe & Ramamoorthy, 2020) | $22.91 \pm 0.19$ | 70M | $128 \times 128$ |
| VideoGPT (Yan et al., 2021) | $24.69 \pm 0.3$ | 200M | $128 \times 128$ |
| TGANv2 (Saito et al., 2020) | $26.60 \pm 0.47$ | 200M | $64 \times 64$ |
| DVD-GAN (Clark et al., 2019) | $\mathbf{32.97} \pm 1.7$ | | $128 \times 128$ |
| RaMViD ($p_U = 0.5$) | $20.84 \pm 0.08$ | 308M | $64 \times 64$ |
| RaMViD ($p_U = 0.75$) | $21.71 \pm 0.21$ | 308M | $64 \times 64$ |

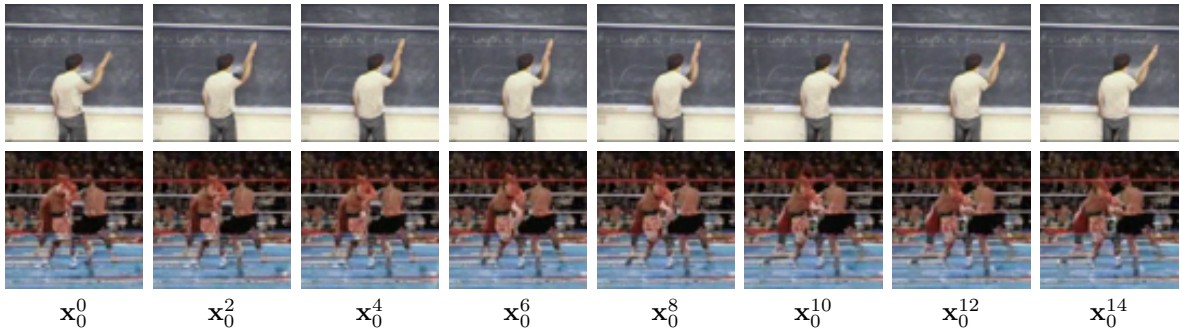

$$\mathbf{x}_0^0 \qquad \mathbf{x}_0^2 \qquad \mathbf{x}_0^4 \qquad \mathbf{x}_0^6 \qquad \mathbf{x}_0^8 \qquad \mathbf{x}_0^{10} \qquad \mathbf{x}_0^{12} \qquad \mathbf{x}_0^{14}$$

Figure 5: Unconditional generation on the UCF-101 dataset. The first generation does not have much movement and is generated very realistically. In the second video, we the background is generated properly, but we see that the fast-moving people are unrealistically deformed.

## 4.4 UCF-101

We train RaMViD on UCF-101 with the same setting as used for Kinetics-600 but for 450,000 iterations. Table 5 shows that our model achieves competitive performance on unconditional video generation, although it does not reach state-of-the-art. The trained models can successfully generate scenes with a static background and a human performing an action in the foreground, consistent with the training dataset (see Figs. 5 and 13). However, the actions are not always coherent and moving objects can deform over time. Note that UCF-101 is a very small dataset given its complexity. Therefore we do observe some overfitting. Since for each action we only have around 25 different settings, our model does not learn to combine those but generates very similar videos to the training set. Due to the characteristics of this dataset we think with more extensive hyperparameter tuning, one can achieve better results with RaMViD in unconditional generation. But our focus does not lie on this.

## 4.5 Autoregressive sampling

While we train our models only on 16 (Kinetics-600) or 20 (BAIR) frames, it is still possible to sample longer sequences autoregressively. By conditioning on the latest sampled frames, one can sample the next sequence and therefore generate arbitrarily long videos. In Fig. 6, we show examples of this autoregressive sampling with RaMViD ($p_U = 0.25$) trained on Kinetics-600. However, we found that this is rather challenging because, at each autoregressive step, the quality of the generated sequence slightly deteriorates. This amplifies over time, often resulting in poor quality after about 30 frames.

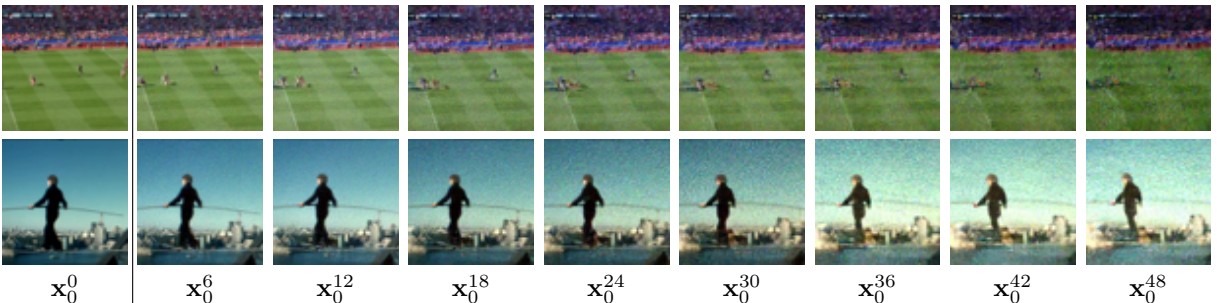

$$\mathbf{x}_0^0 \qquad \mathbf{x}_0^6 \qquad \mathbf{x}_0^{12} \qquad \mathbf{x}_0^{18} \qquad \mathbf{x}_0^{24} \qquad \mathbf{x}_0^{30} \qquad \mathbf{x}_0^{36} \qquad \mathbf{x}_0^{42} \qquad \mathbf{x}_0^{48}$$

Figure 6: Autoregressive prediction of 45 frames conditioned on 5 frames with RaMViD ($p_U = 0.25$) trained on Kinetics-600.

## 5 Conclusion

We have shown that diffusion models, which have been demonstrated to be remarkably powerful for image generation, can be extended to videos and used for several video completion tasks. The way we introduce conditioning information is novel, simple, and does not require any major modification to the architecture of existing diffusion models, but it is nonetheless surprisingly effective. Although the proposed method targets conditional video generation, we also introduce an alternative masking schedule in an attempt to improve the unconditional generation performance without sacrificing performance on conditional generation tasks.

Since we have observed varying performance in different tasks using different masking schemes, an interesting direction for future research is to investigate which masking schedules are more suitable for each task. It would also be interesting to explore in future work whether our conditioning technique is also effective for completion on other data domains. Finally, the focus of this work has been on the diffusion-based algorithm for videos rather than on optimizing the quality of each frame. It has been shown in concurrent works that including super-resolution modules helps create high-resolution videos. Adding a super-resolution module to RaMViD would be a relevant direction for future work.

### Broader impact statement

Generative models have been a cornerstone of AI research for many years. While some of these models have been used for the benefit of society e.g., in wildlife conservation, they can likewise be used with malicious intent such as for creating deepfakes. Similarly to other generative models, the capabilities of our approach for filling and predicting videos can be used for the benefit of all as well as unfortunately the opposite. In this work, we can not see any specific negative impact beyond the general possibility of malicious users of AI algorithms. In addition, our models require large video datasets for training. For some of these curated datasets, the underlying distribution of samples across various groups might not be uniform and before deploying any machine learning model trained on these datasets these aspects need to be carefully evaluated.

### Acknowledgments

This project was enabled by the Berzelius cluster at the Swedish National Supercomputer Center (NSC). We thank our anonymous reviewers for their constructive feedback.

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

# A    Implementation details

Fig. 7 presents a sketch of RaMViD's architecture. Thanks to the way we introduce conditioning frames, the architecture does not need to be different from the one in unconditional models.

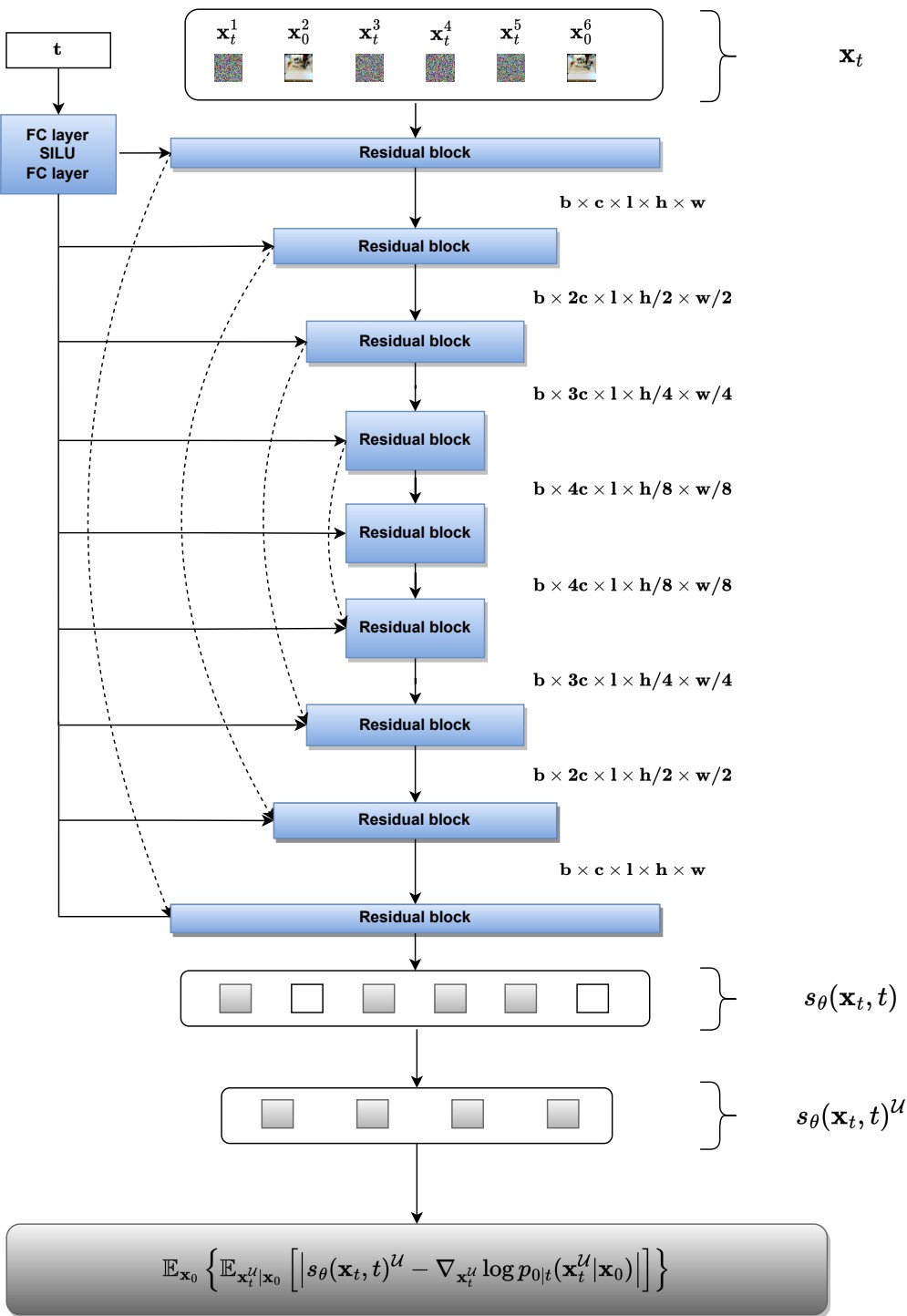

Figure 7: Sketch of our method. In the last step we only compute the loss with respect to the frames that were corrupted with noise. The number of channels **c** is 128, and **l** is the video length.

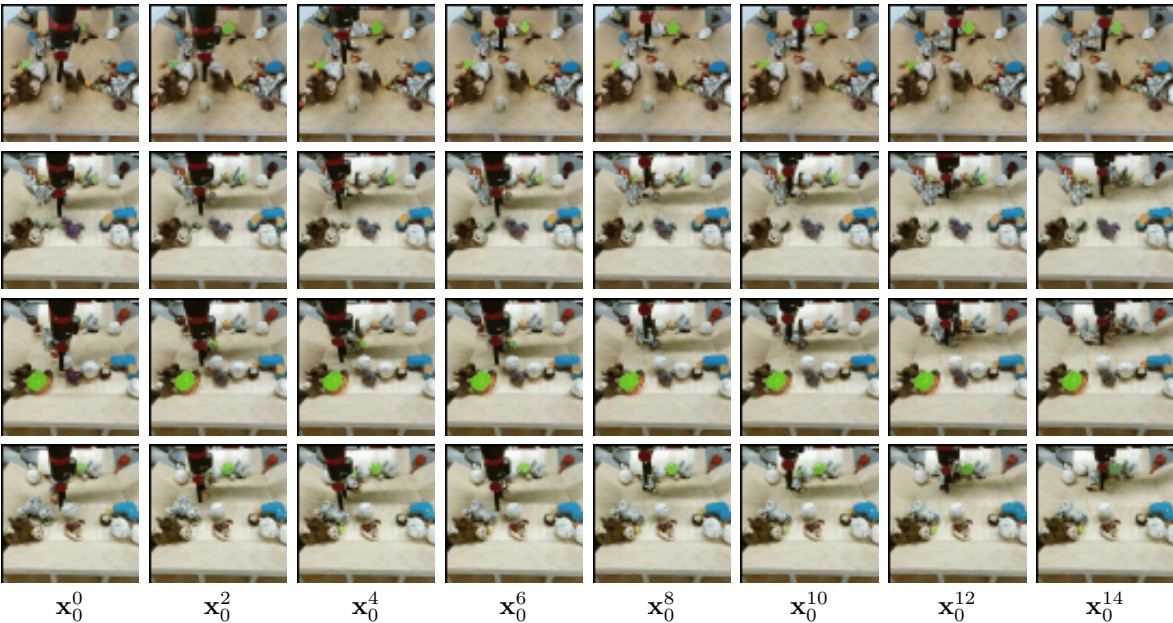

$\mathbf{x}_0^0 \qquad \mathbf{x}_0^2 \qquad \mathbf{x}_0^4 \qquad \mathbf{x}_0^6 \qquad \mathbf{x}_0^8 \qquad \mathbf{x}_0^{10} \qquad \mathbf{x}_0^{12} \qquad \mathbf{x}_0^{14}$

Figure 8: Unconditional generation on the BAIR dataset sampled from RaMViD for $p_U = 0$ (first row) until $p_U = 0.75$ (last row). Due to the low complexity of the dataset, we can generate reasonable unconditional videos even with $p_U = 0$. However, the quality of details increases with increasing $p_U$.

As mentioned, we use the linear noise schedule and the score-based ("simple") objective for all experiments. All models are trained with 1000 diffusion steps, for sampling we used 750 steps on BAIR, and 500 on Kinetics-600 and UCF-101.

As mentioned in Section 4, we use the code base from Nichol & Dhariwal (2021) (MIT license) and their proposed architecture, except that we use $3 \times 3 \times 3$ convolution kernels. In the encoder, we downsample only the spatial dimensions down to $8 \times 8$ in three steps. We use 128 channels for the first block and increase it by 128 for each downsampling step. As mentioned in Section 4, we use multi-head self-attention at the resolutions 16 and 8, each with 4 attention heads. For sampling, we found it to be more beneficial to sample from the exponential moving average (EMA) of the weights (Nichol & Dhariwal, 2021). We set the EMA rate to 0.9999.

## B    Additional results

### B.1    Results on BAIR

Fig. 8 shows that all of the models are also able to do unconditional video generation (even RaMViD ($p_U = 0$), we assume that this is due to the low diversity of the dataset). Qualitatively, we can see in Fig. 8 that videos generated by models with higher $p_U$ are better in generating details. While all models can generate the moving robot arm, only the models with $p_U \geq 0.25$ can properly generate the different objects in the box. However, we have no quantitative results on unconditional generation on BAIR.

### B.2    Results on Kinetics-600

Kinetics-600 in practice appears to be the most difficult dataset among those considered here. While our results are state-of-the-art (see Table 3), we do observe failure cases. One of the most common failure cases is fast movement. In that case we often see a deformation of the moving object (see Fig. 9).

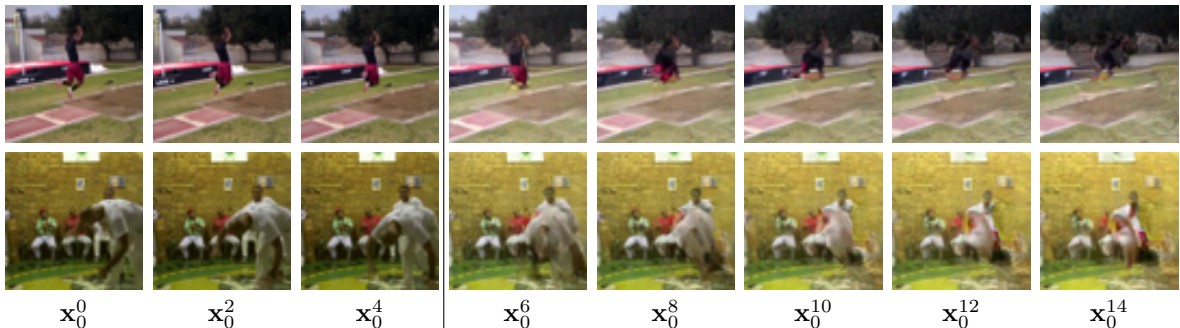

$\mathbf{x}_0^0$ $\quad$ $\mathbf{x}_0^2$ $\quad$ $\mathbf{x}_0^4$ $\quad$ $\mathbf{x}_0^6$ $\quad$ $\mathbf{x}_0^8$ $\quad$ $\mathbf{x}_0^{10}$ $\quad$ $\mathbf{x}_0^{12}$ $\quad$ $\mathbf{x}_0^{14}$

Figure 9: Prediction of 11 frames given the first 5 frames on Kinetics-600 when conditioning on fast-moving objects. We can see that the background does get preserved well, while the object itself gets unrealistically deformed.

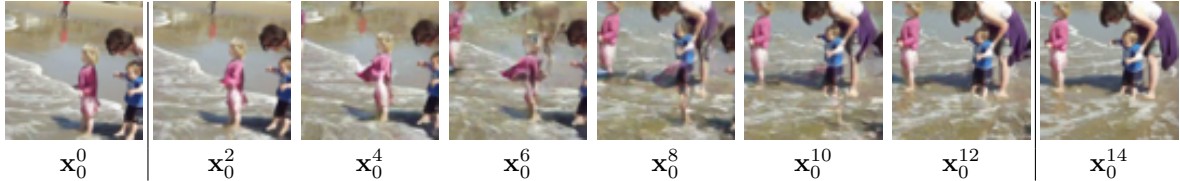

$\mathbf{x}_0^0$ $\quad$ $\mathbf{x}_0^2$ $\quad$ $\mathbf{x}_0^4$ $\quad$ $\mathbf{x}_0^6$ $\quad$ $\mathbf{x}_0^8$ $\quad$ $\mathbf{x}_0^{10}$ $\quad$ $\mathbf{x}_0^{12}$ $\quad$ $\mathbf{x}_0^{14}$

Figure 10: Infilling on Kinetics. The conditioned frames are $\mathcal{C} = \{0, 1, 14, 15\}$. The people in frames 0 and 1 are placed quite differently than in 14 and 15. The model is not able to generate the necessary camera movement and does simply interpolate between the frames.

Fast camera movement can also be a problem for infilling. If an object is placed very different between the first and last frames, the model does not generate a harmonized movement but makes the object disappear in the first and appear in the last frames Fig. 10.

## C   Qualitative comparison

The main qualitative improvement of RaMViD compared to other methods is the decrease in occurence of deformed objects in the predictions on Kinetics-600 (see Fig. 15). While the predictions of several other methods suffer from object deformations, RaMViD only suffers from this with fast-moving objects. On BAIR, on the other hand, our qualitative results are similar to the ones of recent methods, although RaMViD can predict details in the interactions in more detail (see Fig. 14).

## D   Compute

Each model is trained on 8 NVIDIA A100 GPUs with 40 GB of memory. The models on BAIR are trained with a batch size of 32 and a micro-batch size of 16 for 250k iterations (~3 days). All other models on Kinetics-600 and UCF-101 are trained with a batch size of 64 and micro-batch size of 16. The models are trained for 500k iterations on Kinetics-600 (~10 days) and for 450k iterations on UCF-101 (~9 days).

## E   Datasets

The videos in all datasets have more frames than we train on. Therefore we choose random sub-sequences of the desired length during training.

**BAIR robot pushing.**   The BAIR robot pushing dataset can be used under an MIT license. We use the low resolution dataset ($64 \times 64$). Since the data is already in the correct size, no prepossessing is necessary. For evaluation we predict one sequence for each of the 256 test videos and compare the FVD to the ground

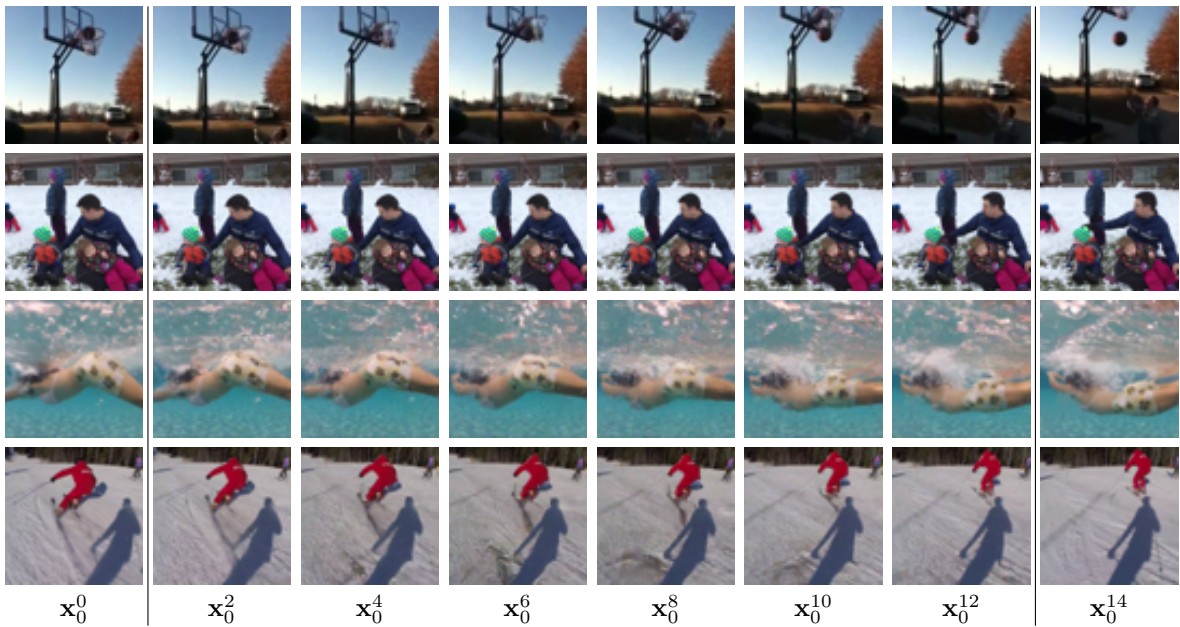

$\mathbf{x}_0^0$  $\mathbf{x}_0^2$  $\mathbf{x}_0^4$  $\mathbf{x}_0^6$  $\mathbf{x}_0^8$  $\mathbf{x}_0^{10}$  $\mathbf{x}_0^{12}$  $\mathbf{x}_0^{14}$

Figure 11: Video infilling ($\mathcal{C} = \{0, 1, 14, 15\}$) on Kinetics-600 with RaMViD ($p_U = 0.25$).

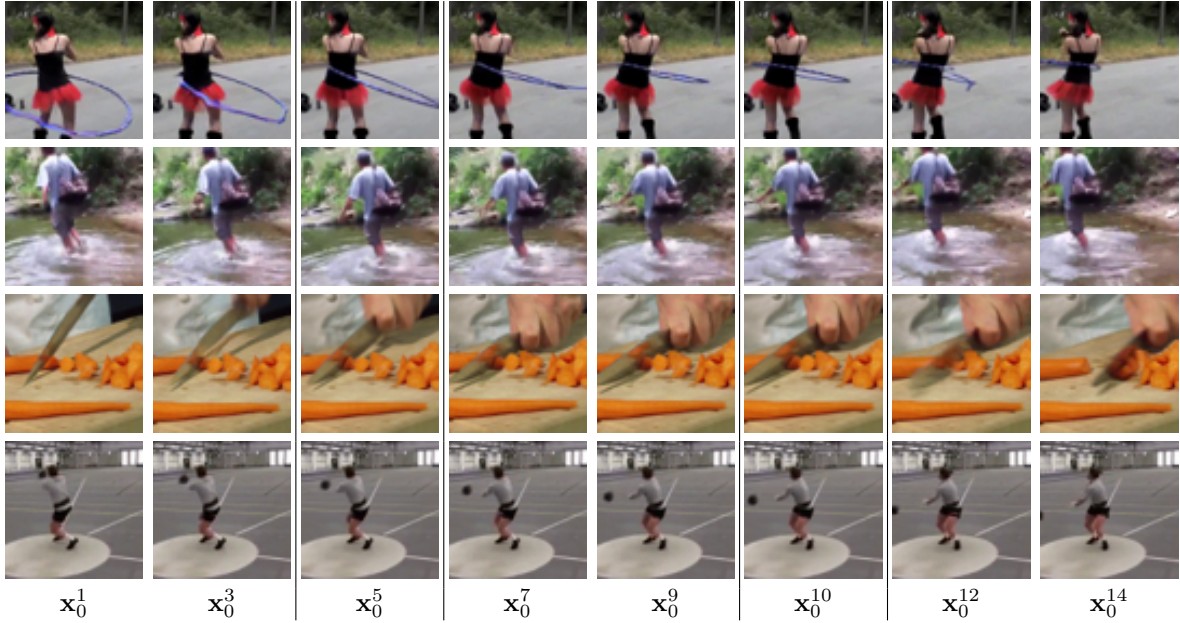

$\mathbf{x}_0^1$  $\mathbf{x}_0^3$  $\mathbf{x}_0^5$  $\mathbf{x}_0^7$  $\mathbf{x}_0^9$  $\mathbf{x}_0^{10}$  $\mathbf{x}_0^{12}$  $\mathbf{x}_0^{14}$

Figure 12: Video completion ($\mathcal{C} = \{0, 5, 10, 15\}$) on Kinetics-600 with RaMViD ($p_U = 0.25$).

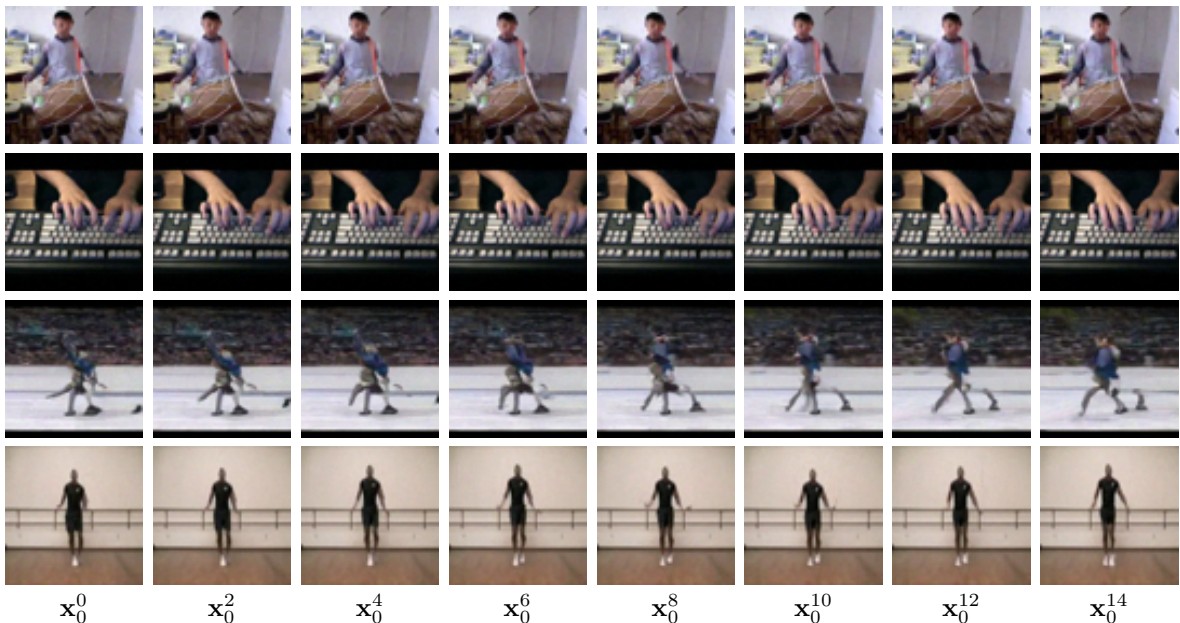

$$\mathbf{x}_0^0 \qquad \mathbf{x}_0^2 \qquad \mathbf{x}_0^4 \qquad \mathbf{x}_0^6 \qquad \mathbf{x}_0^8 \qquad \mathbf{x}_0^{10} \qquad \mathbf{x}_0^{12} \qquad \mathbf{x}_0^{14}$$

Figure 13: Qualitative results of unconditional generation on UCF-101. Scenes with less movement are generated well, but are often close to the training set.

truth. To get a proper evaluation score, we do this 100 times and the final FVD score is the average over all 100 runs. We train on a sequence length of 20.

**Kinetics-600.** The Kinetics-600 dataset has a Creative Commons Attribution 4.0 International License. The videos have different resolutions, which is why we reshape and center crop them to a $64 \times 64$ resolution. For evaluation we take 50,000 videos from the test set and predict a sequence for each of the videos. We then compute the statistics for the ground truth and the predicted videos to obtain the FVD score. We train on a sequence length of 16.

**UCF-101.** We could not find a license for the UCF-101 dataset. The original frames have a resolution of $160 \times 120$, therefore we resize and center crop the videos to a $64 \times 64$ resolution. We train on the entire dataset of 13,320 videos. To evaluate the generative performance, we sample 10000 videos unconditionally and compute the Inception Score (IS).[6] This is repeated three times.

## F  Sampling Speed

Since sampling 10,000 videos takes about 9 hours with the large model trained on Kinetics-600 and UCF-101, and 7 hours with the smaller model trained on BAIR using 500 sampling steps, it is crucial to know how many sampling steps are necessary to achieve satisfactory performance. We found that by using only 250 sampling steps for RaMViD (p = 0.25) trained on BAIR, our results drop significantly (to 101.54). However, when using 500 sampling steps, we achieve an FVD of 85.07 which is very similar to using 750 steps (84.20). We observe the same behaviour on Kinetics-600, where we achieve an FVD of 14 / 16 with 750 / 500 sampling steps but only an FVD of 49 with 250 steps. Therefore, when using the DDPM sampler with RaMViD, we recommend using a minimum of 500 sampling steps for video generation.

---

[6]https://github.com/pfnet-research/tgan2

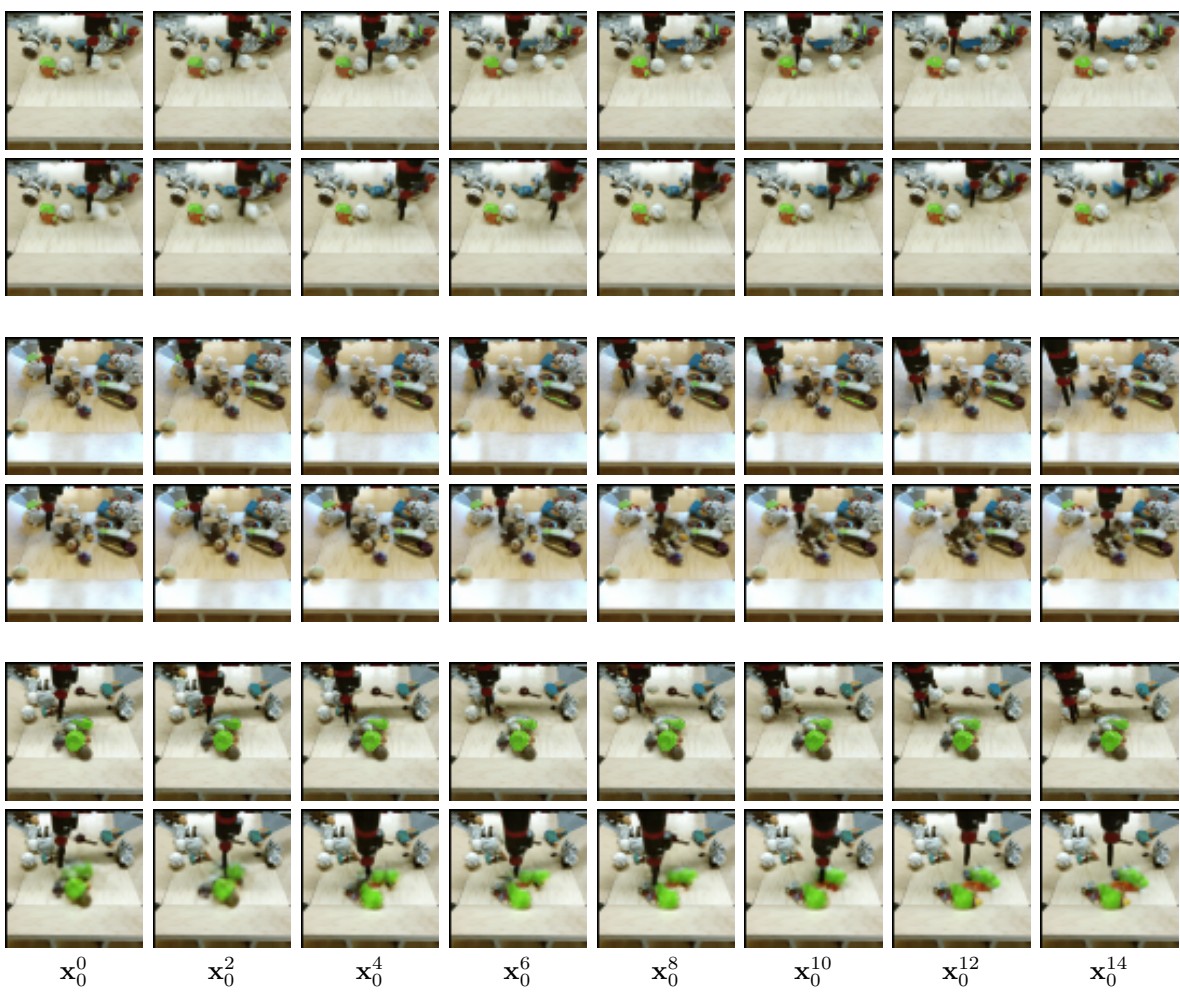

$$\mathbf{x}_0^0 \qquad \mathbf{x}_0^2 \qquad \mathbf{x}_0^4 \qquad \mathbf{x}_0^6 \qquad \mathbf{x}_0^8 \qquad \mathbf{x}_0^{10} \qquad \mathbf{x}_0^{12} \qquad \mathbf{x}_0^{14}$$

Figure 14: A comparison between RaMViD ($p_U = 0.25, K = 8$) and VideoGPT (Yan et al., 2021). The first row shows our predictions and the second VideoGPT respectively. Visually, the difference is not significant, but we found that VideoGPT produces artefacts, especially when there are interacting objects.

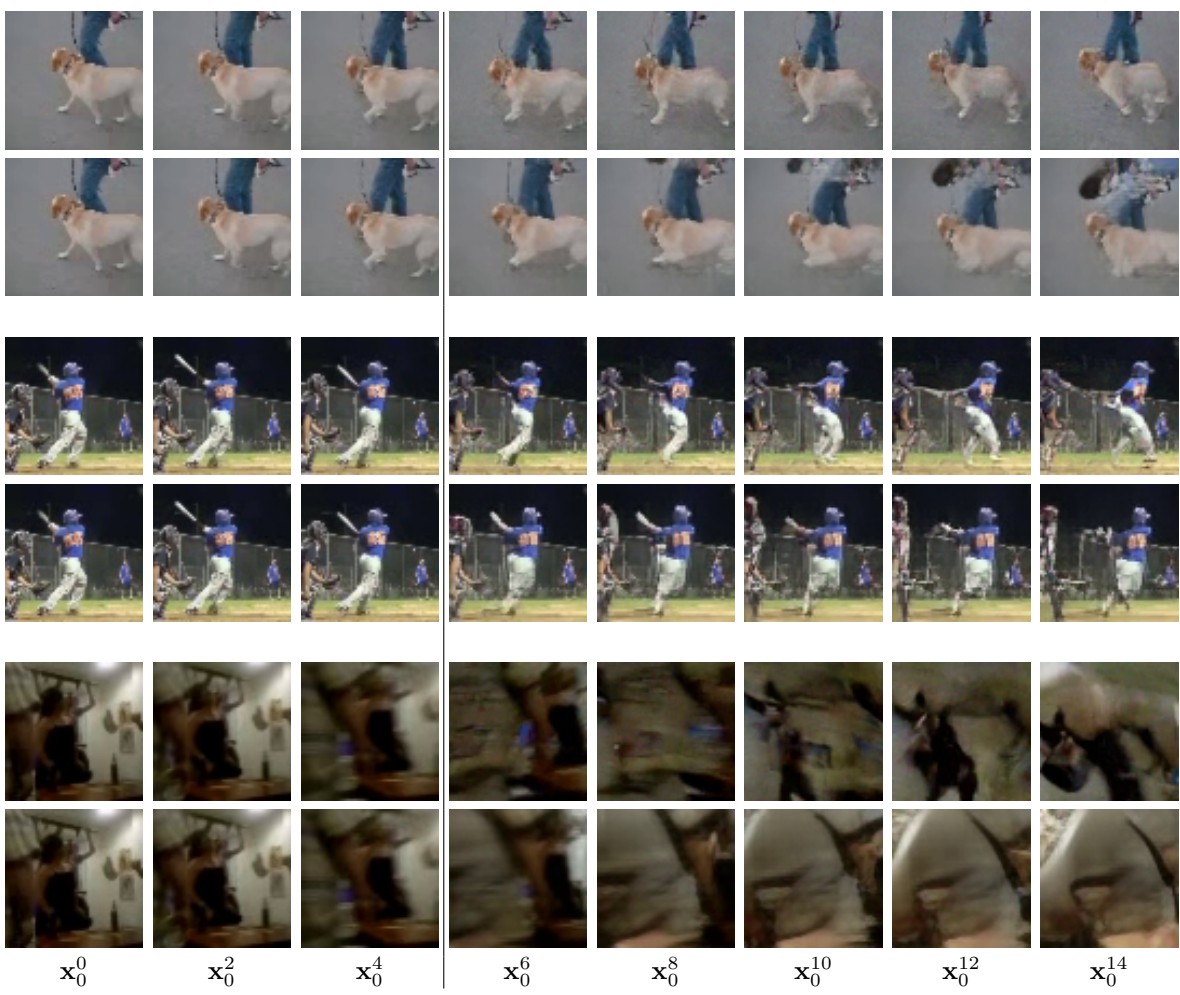

$\mathbf{x}_0^0 \qquad \mathbf{x}_0^2 \qquad \mathbf{x}_0^4 \qquad \mathbf{x}_0^6 \qquad \mathbf{x}_0^8 \qquad \mathbf{x}_0^{10} \qquad \mathbf{x}_0^{12} \qquad \mathbf{x}_0^{14}$

Figure 15: Predicted videos on Kinetics-600 from RaMViD ($p_U = 0.25$) in the top row, and CCVS in the bottom row. We found that our model is better in modelling motion, whereas for more static videos, the difference is visually not significant. Nevertheless, we also found that both models perform equally bad for fast object/camera movement.

## G    Concurrent work

As mentioned in Section 2, three concurrent works on diffusion models for videos were recently made public. Only Ho et al. (2022) and Voleti et al. (2022) consider similar tasks as we do. Ho et al. (2022) appears to outperform RaMViD on unconditional video generation on UCF-101, which is not surprising, as we train with the mixed method and therefore the models are mostly trained for conditional generation. Voleti et al. (2022) evaluate their method on BAIR with the same procedure we used, and the results reported in their publication suggest that RaMViD outperforms their proposed method.

