# OpenReview forum: "Diffusion Models for Video Prediction and Infilling"
_TMLR — Accepted by TMLR_

### Review · Reviewer_cFap · 2022-08-24

**Summary Of Contributions:**

This paper proposes the extension of diffusion models to the domain of video prediction, generation, and infilling. The proposed method works for both unconditional video generation, as well as conditional video generation (i.e., conditioned on a set of initial frames). In order to achieve this, the paper introduces a new way to condition diffusion models on video information, by masking out frames at random. This enables their model to be jointly trained on conditional and unconditional examples, and perform a variety of tasks. The proposed method achieves state-of-the-art on several video benchmarks as measured by automatic metrics on video quality (FVD and IS).


**Broader Impact Concerns:**

The primary concern of the paper is on adversarial and unethical deployment of generative models, e.g., for creating deepfakes. This is well addressed by Section 6 of the paper.

**Requested Changes:**

In general, I believe that the results and modeling from this paper are valuable contributions for the video generation and generative modeling community. The proposed model is simple and effective, trained on modest resources to achieve SOTA. However, I think that the paper would be significantly improved if the evaluation procedure in the paper was more comprehensive and robust:

- As mentioned in the “weaknesses section”: It would be useful to include measures of efficiency and capacity of each model. Does the proposed model improve upon baselines simply because it has greater capacity, or is it due to diffusion-based training?
- More ablation analysis on the masking strategy (which appears to be one of the primary contributions of the paper) is essential. In particular, running ablation experiments over $K$ (number of conditioning frames), the random masking strategy (uniform vs. non-uniform), masking chunks rather than individual frames.

**Strengths And Weaknesses:**

# Strengths
The proposed model is conceptually simple -- the model itself is a modification of [1] to use 3D convolutional kernels instead of 2D kernels.
- The paper introduces a simple and careful masking strategy to enable ease of training on unconditional and conditional examples. This enables the proposed model to achieve SOTA on several video generation benchmarks by training with modest resources.
- The paper is generally well written and easy to follow.


# Weaknesses

The evaluation and analysis of the model is fairly insufficient, with few ablative studies. There are several aspects that could be significantly improved:
- It would be useful to include other diffusion models into the evaluation tables. In particular, all of the other baseline models in Tables 1, 3, and 5 are not diffusion models, and including some of the diffusion models (from the “concurrent work” paragraph of section 2) would be useful for an apples-to-apples comparison. Even if the results are considered concurrent for the purposes of SOTA, it would still be useful to provide the reader with a more comprehensive view of the diffusion modeling landscape.
- It would be useful to include some measure of computational efficiency (in the form of FLOPS, inference throughput, training throughput) to compare the proposed model against baselines. It would be especially useful to include an efficiency comparison between diffusion based models and autoregressive models.
- Related to the above point, it would also be useful to include the parameter count of each model being compared. In some prior papers (e.g., [2]), it appears that generative diffusion models achieve SOTA with a much lower parameter count compared to traditional GAN or autoregressive based approaches. Do we observe a similar trend here, for video generation?
- The paper does not contain many ablations. It would be useful to conduct a study on the effect of the masking procedure against results (the only ablation on this is on the probability of training on unconditional examples). For example, it might be worthwhile to study masking strategies that do not sample uniformly at random -- would masking “chunks” rather than individual frames be helpful? Or would masking more frames at the beginning of the video be helpful/harmful as compared to masking at random?


# References
[1] Nichol, Alexander Quinn, and Prafulla Dhariwal. "Improved denoising diffusion probabilistic models." International Conference on Machine Learning. PMLR, 2021.

[2] Saharia, Chitwan, et al. "Photorealistic Text-to-Image Diffusion Models with Deep Language Understanding." arXiv preprint arXiv:2205.11487 (2022).

---

> ### Author Response · Authors · 2022-09-26
> **Addressing Requested Changes**
>
> First, we would like to thank the reviewer for their helpful and detailed review.
>
> _**As mentioned in the “weaknesses section”: It would be useful to include measures of efficiency and capacity of each model. Does the proposed model improve upon baselines simply because it has greater capacity, or is it due to diffusion-based training?**_
> RaMViD trained on BAIR has 235M parameters and trained on Kinetics-600 308M, while the largest concurrent work has 373M parameters (for both datasets). In general, we found that our model is slightly smaller than recent SOTA methods.  For UCF-101 however, we found that our model is one of the largest. We have also updated the tables with the capacity of the models for easy comparison. We have obtained this information either by contacting the authors or directly from their work.
>
> _**More ablation analysis on the masking strategy (which appears to be one of the primary contributions of the paper) is essential. In particular, running ablation experiments over  (number of conditioning frames), the random masking strategy (uniform vs. non-uniform), masking chunks rather than individual frames.**_
> We have added ablation studies on BAIR about the maximal number of masked frames (K) and about fixed conditional frames. Due to limited compute, we were unable to complete ablation studies on Kinetics-600 or UCF-101, but we will include these results in the future.

---

> > ### Comment · Reviewer_cFap · 2022-10-13
> > **Reply**
> >
> > Thanks for this - the numbers are very helpful for building a good mental model of video diffusion models.

---

### Review · Reviewer_LFdv · 2022-08-29

**Summary Of Contributions:**

The paper proposed an approach for video infilling, prediction, and generation, RaMViD, based on 3D convolutional neural networks and diffusion generative models. Given observed conditional frames, 3D convolutional model learns to approximate the score function of the unknown frames. During training, the observed conditional frames are randomly sampled from each training sequence and the model is randomly switched between being trained to approximate conditional score functions and unconditional score functions with a certain probability. The work evalutes the model on various tasks including video prediction and infilling and unconditional generation using standard benchmarks and datasets, including BAIR, Kinetics-600 and UCF-101. The proposed approach demonstrates state-of-the-arts results on various tasks.

**Broader Impact Concerns:**

The broader impacts of the work has been fully discussed. The reviewer has no concern on this part.

**Requested Changes:**

1. The work need to include comparison of qualitative samples between RaMViD and baseline approaches as well as detailed discussion on comparison results.

2. Equation 8 and Equation 9 can be confusing. In Equation 8, it is not clear what the second parenthesis in $s_\theta (x_t, t)(x_t^U) $ means. Apart from the reversed diffusion process in Equation 9, the paper should also clearly presents the equation of the forward diffusion process (from data to noise) in the conditional setting.

3. The numbers in Table 3 are not reported in the same format as the previous models, can the author explain why?

4. Given the stochastic nature of diffusion models, the paper should report mean and standard deviations of results of multiple runs.


**Strengths And Weaknesses:**

Strength:

1. The experiment results of the proposed approach is impressive. It outperforms existing baseline models by a significant margin on various tasks including on standard benchmarks datasets, including. The work also did extensive study on different settings and hyperparameters, including the conditioning frame numbers and propability of doing unconditional training.

2. The use of diffusion model for video infilling and prediction is an interesting application of diffusion model. It is of great importance and relevance to both generative modeling and video prediction communities.

3. The work is easy to follow and enjoyable to read. RamViD is well presented with the help of pseudocode.

4. It should also be appreciated that the work also discusses the RaMViD's challenges when handling fast moving objects on Kinetics-600 and shows unconditional generation results on UCF-101 where RaMViD underperform existing models.

Weakness:

1. Despite impressive quantitaive experiment results, there're no comparison of qualitative sampels between RaMViD and other approaches. Comparisons on qualitative samples could help better demonstrates the model's strengths and weakness and provide more insights into the model.

2. There are several presentation issues. Please see requested changes for more details.

---

> ### Author Response · Authors · 2022-09-26
> **Addressing Requested Changes**
>
> First, we would like to thank the reviewer for their helpful and detailed review.
>
> _**The work need to include comparison of qualitative samples between RaMViD and baseline approaches as well as detailed discussion on comparison results.**_
> The main qualitative improvement of RaMViD compared to other methods is the decrease in deformed objects in the predictions on Kinetics-600. Many methods do suffer from predictions in which objects get unrealistically deformed while RaMViD only suffers from this with fast-moving objects. On BAIR, on the other hand, our qualitative results were similar to the ones of recent methods, but RaMViD was able to predict details in the interactions in more detail. We found that a qualitative comparison in the paper itself (i.e. only with still frames from the videos) is not very informative. Therefore, we would encourage readers to check out our website as well as published sites from concurrent work.
>
> _**Equation 8 and Equation 9 can be confusing. In Equation 8, it is not clear what the second parenthesis in means. Apart from the reversed diffusion process in Equation 9, the paper should also clearly presents the equation of the forward diffusion process (from data to noise) in the conditional setting.**_
> We updated our notation in Equation 8 and added the Equation for the forward process as well
>
> _**The numbers in Table 3 are not reported in the same format as the previous models, can the author explain why?**_
> The authors of CCVS (from which we took the numbers) have rounded the results simply for easy comparison to prior work. However, we preferred to have some higher precision, which is why we did report it in a different format.
>
> _**Given the stochastic nature of diffusion models, the paper should report mean and standard deviations of results of multiple runs.**_
> On BAIR it was rather unusual to report multiple runs, which is why we also did not consider doing this. On UCF-101 it is common practice to report the mean and standard deviation of multiple batches (each batch has 10000 videos) which we did. For Kinetics we agree that reporting statistics from multiple runs would be valuable and we have started retraining our best model (RaMViD (p = 0.25)) to obtain the mean and standard deviation.  Unfortunately, our computational resources are rather scarce at this moment (due to the ICLR deadline), which is why we might not be able to report new values until the end of the discussion period. We hope that our state-of-the-art results with the same model on two datasets hint at the robustness and stability of our method.

---

> > ### Author Response · Authors · 2022-09-30
> > **Mean and standard deviations of results of multiple runs**
> >
> > Unfortunately, the models on Kinetics-600 are still running and probably won’t be done in time. However, in order to give at least some intuition about the mean and standard deviation of RaMViD, we have trained more models on BAIR with p = 0.25. We used 500 steps for sampling and the mean and standard deviation of three training runs is 86.3 +/- 1.8. This shows, that the results of RaMViD are quite stable.

---

> > > ### Author Response · Authors · 2022-10-11
> > > **Mean and standard deviations of results of multiple runs on Kinetics-600**
> > >
> > > Only one other model on Kinetics-600 has finished training on time. The mean and standard deviation from the two runs of RaMViD (p=0.25) on Kinetics-600 is 17.53 +/- 1.07. We know that for a proper estimate of the mean and standard deviation more runs are needed, but we believe that these results, combined with our estimates of the mean and standard deviation on BAIR, indicate the robustness of our method.

---

### Review · Reviewer_XQ2a · 2022-09-14

**Summary Of Contributions:**

The paper introduces a diffusion model for video prediction, where both an "unconditional model" and a conditional model are trained jointly. The method achieves better performance on video prediction compared to some baselines in terms of FVD and IS.

**Broader Impact Concerns:**

No.

**Requested Changes:**

- Change the claims regarding Palette, which is not correct regarding inpainting.
- Add some discussion about the speed/latency of the predictions, the sampler being used, and an ablation study about the trade-off between prediction quality versus speed.

**Strengths And Weaknesses:**

Strengths:
- One of the earlier (concurrent) works that applies diffusion model to video prediction and infilling.
- The empirical results are solid and outperform traditional methods.

Weaknesses:
- The technical novelty is a bit limited, as the approach is proposed in the Palette paper in inpainting. Despite what is claimed in the paper here, the inpainting method proposed by Palette also do not increase the dimensions of the model input, where random Gaussian noise are added to masked regions as well for inputs.
- There is less discussion about what sampler is used in these tasks, which is very critical for time-sensitive applications. While it is acceptable to be slower than GANs, it is still helpful to discuss the time spent in these tasks and perform a small ablation study on top of it.

---

> ### Author Response · Authors · 2022-09-26
> **Addressing Requested Changes**
>
> First, we would like to thank the reviewer for their helpful and detailed review.
>
> _**Change the claims regarding Palette, which is not correct regarding inpainting.**_
> We are sorry for being unclear about this. [1] use zero padding. [2,3,4] mask the image (i.e. it stays at the same dimension) and then concatenate it with the noisy image, so the model input has 6 channels. Since we do not rely on concatenation, the input of RaMViD has only 3 channels.  We have updated the section in our new version
>
> [1] Yusuke Tashiro, Jiaming Song, Yang Song, and Stefano Ermon. CSDI: Conditional score-based diffusion models for probabilistic time series imputation, 2021.
> [2] Chitwan Saharia, William Chan, Huiwen Chang, Chris A. Lee, Jonathan Ho, Tim Salimans, David J. Fleet, and Mohammad Norouzi. Palette: Image-to-image diffusion models, 2021a.
> [3] Georgios Batzolis, Jan Stanczuk, Carola-Bibiane Schönlieb, and Christian Etmann. Conditional image generation with score-based diffusion models, 2021.
> [4] Chitwan Saharia, Jonathan Ho, William Chan, Tim Salimans, David J. Fleet, and  Mohammad Norouzi. Image super-resolution via iterative refinement, 2021b.
>
>
> _**Add some discussion about the speed/latency of the predictions, the sampler being used, and an ablation study about the trade-off between prediction quality versus speed.**_
> We use the DDPM sampler. Since we sample all 16 frames at once, sampling takes quite a long time. Generating 10000 videos with the larger model for Kinetics-600 or UCF-101 takes about 9 hours when using 500 sampling steps. Generating the same amount of videos with the same amount of sampling steps takes around 7 hours for the smaller model used for BAIR. When using only 250 sampling steps for RaMViD (p = 0.25) trained on BAIR, our results drop significantly to 101.54. However, when using 500 sampling steps, we achieve an FVD of 85.07 which is very similar to using 750 steps (84.20). We observe the same behaviour on Kinetics-600, where we achieved an FVD of 14 / 16 with 750 / 500 sampling steps but only an FVD of 49 with 250 steps.
> We have added this information to the appendix of the paper (Section E: Sampling Speed).

---

### Author Response · Authors · 2022-09-26
**Limited compute**

We would like to thank all reviewers for their helpful feedback. We are sorry for responding so late, but unfortunately, we currently (due to the ICLR deadline) do not have enough compute to run all requested experiments. However, we were able to train some more models and run some more samplings and we hope that these are sufficient. Some models are still queuing, but we are unsure if they will be finished by the end of the discussion period.

---

### Decision · Action_Editors · 2022-10-24

**Recommendation:** Accept as is

**Comment:**

Overall, the paper presents a simple diffusion-based video prediction method with strong performance demonstrated in multiple settings. The paper is written clearly and authors have modified the manuscript in response to reviewer concerns. This includes adding discussion on sampling speed, updating notation for clarity, adding qualitative examples, adding model size discussion, and further examination of concurrent work. The AE believes the claims in this work are well-supported and there exists an audience of TMLR readers who would be interested in the method, analysis, and provided codebase.

**Audience:**

Reviewers agree that this work presents a simple and early effort to adapt 2D image diffusion models to video prediction, generation, and infilling. The audience for this work seems clear. Quoting reviewer LFdv -- "It is of great importance and relevance to both generative modeling and video prediction communities."

**Claims And Evidence:**

Reviewers cFap and XQ2a both provide positive support -- citing extensive experiments and strong performance on common datasets as well as availability of code. Reviewer LFdv recognizes these same points, citing "impressive" performance and "extensive study on different settings and hyperparameters"; however, their recommendation towards rejection is motivated by a lack of qualitative examples in the paper itself and lack of standard deviations over multiple runs in evaluation.

Authors have provided qualitative examples on an associated webpage as videos and still-frame sequence examples in the appendix. They have also presented standard deviations for some limited settings rather due to computational constraints during the response period. Combined with standard deviation in their other experiments, these suggest the results are sufficiently stable to support the claims. Authors are encouraged to include these numbers in the camera ready.